# Disentangling the roles of dimensionality and cell classes in neural computations

A. M. Dubreuil[1], A. Valente[1], F. Mastrogiuseppe[2], S. Ostojic[1]

*1. Group for Neural Theory, École Normale Supérieure, Paris, France*

*2. Gatsby Computational Neuroscience Unit, UCL, London, UK*

The description of neural computations in the field of neuroscience relies on two competing views: (i) a classical single-cell view that relates the activity of individual neurons to sensory or behavioural variables, and focuses on how different cell classes map onto computations; (ii) a more recent population view that instead characterises computations in terms of collective neural trajectories, and focuses on the dimensionality of these trajectories as animals perform tasks. How the two key concepts of cell classes and low-dimensional trajectories interact to shape neural computations is however currently not understood. Here we address this question by combining machine-learning tools for training RNNs with reverse-engineering and theoretical analyses of network dynamics. We introduce a novel class of theoretically tractable recurrent networks: low-rank, mixture of Gaussian RNNs. In these networks, the rank of the connectivity controls the dimensionality of the dynamics, while the number of components in the Gaussian mixture corresponds to the number of cell classes. Using back-propagation, we determine the minimum rank and number of cell classes needed to implement neuroscience tasks of increasing complexity. We then exploit mean-field theory to reverse-engineer the obtained solutions and identify the respective roles of dimensionality and cell classes. We show that the rank determines the phase-space available for dynamics that implement input-output mappings, while having multiple cell classes allows networks to flexibly switch between different types of dynamics in the available phase-space. Our results have implications for the analysis of neuroscience experiments and the development of explainable AI.

## 1 Introduction

With recent advances in deep-learning, the novel approach of training and reverse-engineering RNNs on neuroscience tasks has led to insights on the implementation of cognitive processes (see [1] for a review). Reverse-engineering methods have however provided only partial understanding so far, by either focusing on the characterization of neural dynamics and leaving aside the description of learnt connectivity [2], or the converse [3]. Taking advantage of recent theoretical results on low-rank networks [4], we present a reverse-engineering approach that leads to analytically tractable classes of RNNs performing various tasks. Crucially, these classes of models exhibit well defined dimensionality and number of cell classes allowing us to identify the roles of these two properties on neural computation.

## 2 Methods

### 2.1 Theoretical framework: low-rank recurrent networks

We consider recurrent networks of $N$ *tanh* rate units with dynamics defined by:

$$
\begin{aligned}
\frac{d\vec{x}}{dt} &= -\vec{x} + W_{rec}\vec{\phi}(\vec{x}) + W_{in}\vec{u}(t) \\
z(t) &= \vec{W}_{out}^T \vec{\phi}(\vec{x})
\end{aligned}
\tag{1}
$$

where $\vec{u}(t)$ represents inputs to the network and $z$ is a scalar readout modeling the network's output. The directions of the network's inputs and output are defined by the column vectors of $W_{in}$ and $\vec{W}_{out}$, while the recurrent connectivity is given by the rank-$K$ matrix $W_{rec} = \sum_{k=1}^{K} \vec{m}_k \vec{n}_k^T$. For such networks, recurrently generated activity lies in a space of dimension $K$, and is well described by the dynamics of a set of $K$ internal (latent) variables. In the case where connectivity and input vectors are drawn from a joint Gaussian distribution (with entry-independent covariances $\sigma_{ab}$ between entries of vectors $\vec{a}$ and $\vec{b}$), a previous study developed a mean-field theory for the dynamics of the internal variables [1]. As an example, for $K = 2$ and a single input $u(t)$, the dynamics can be described by two internal variables $\kappa_1$ and $\kappa_2$:

$$
\begin{aligned}
\dot{\kappa}_1 &= -\kappa_1 + \tilde{\sigma}_{n_1 m_1}\kappa_1 + \tilde{\sigma}_{n_1 m_2}\kappa_2 + \tilde{\sigma}_{n_1 W_{in}}u(t) \\
\dot{\kappa}_2 &= -\kappa_2 + \tilde{\sigma}_{n_2 m_1}\kappa_1 + \tilde{\sigma}_{n_2 m_2}\kappa_2 + \tilde{\sigma}_{n_2 W_{in}}u(t)
\end{aligned}
\tag{2}
$$

where the functional connectivities

$$\tilde{\sigma}_{ab}(\kappa_1, \kappa_2, u(t)) = \sigma_{ab}\langle\phi'\rangle(\kappa_1, \kappa_2, u(t)) \tag{3}$$

are the product of a structural component $\sigma_{ab}$ and an activity dependent term, namely the population-averaged gain of neurons $\langle\phi'\rangle$.

## 2.2 Theoretical framework: multi-population low-rank recurrent networks

In the present work we extend this framework to describe networks with neurons assigned to $P$ populations, which can be described with connectivity vectors drawn from $P$-mixtures of Gaussians: each neuron $i$ belongs to one of the $P$ populations, and entries of the structure vectors $a_i$ and $b_i$ are drawn from a joint Gaussian distribution with covariance $\sigma_{ab}^{\text{(k)}}$, $k = 1, ..., P$. Thus cell classes are defined in terms of connectivity profiles. In the mean-field theory, the functional connectivities become

$$\tilde{\sigma}_{ab} = \sum_{k=1}^{P} \sigma_{ab}^{\text{(k)}}\langle\phi'\rangle_k \tag{4}$$

where $\langle\rangle_k$ is an average over the entries assigned to the population $k$ of the mixture.

## 2.3 Training and reverse-engineering low-rank RNNs

We consider a series of classical neuroscience tasks and for each we determine a mixture of Gaussians network model, with minimal-rank connectivity and a minimal number of cell classes. Our first step is to train a RNN, in a supervised manner using BPTT and the ADAM algorithm, by optimizing on entries of $\vec{W}_{in}$, $\vec{W}_{out}$, $\{\vec{m}_k\}_k$, $\{\vec{n}_k\}_k$. We thus look for solutions in the restricted space of networks whose connectivity matrix is rank $K$, without imposing well defined Gaussian statistics. We train networks with various values of $K$ and identify the minimal value $K_{min}$ for which a solution can be found. After training, we exploit mean-field theory for low-rank RNNs to reverse-engineer the trained networks. We first relate the internal variables to the task being performed, which allows us to obtain a dynamical system description of the cognitive components at hand. We then extract relevant statistical features of the trained vectors to relate this dynamical system description to the learnt connectivity structure. Guided by this analysis we are able to reconstruct rank-$K_{min}$ RNNs whose connectivity vectors can be described by a $P$-mixture of Gaussians, and to determine the minimal $P$ for which a solution can be found.

## 3 Results

Our approach allowed us to identify two general principles for the roles of dimensionality and cell classes. Below we summarize and illustrate them on a subset of studied tasks, and then exploit them to build networks that perform multiple tasks.

### 3.1 Increasing rank allows to increase the number of internal variables used in the computation

We first consider a perceptual integration task (random dot motion task, RDM, Figure 1A), where a network is exposed to a noisy input signal and is asked to report the sign of the temporally averaged signal. We find that a network with rank $K = 1$ and $P = 1$ population is able to perform this task (Figure 1B). The internal variable $\kappa$ is easily interpreted in terms of the computation performed by the network: it integrates the input signal before converging to either one of two fixed-points encoding the positive/negative decision (Figure 1C). This internal variable closely matches accumulation of evidence in drift-diffusion models that have been proposed to model this type of perceptual integration tasks.

We next consider a parametric working memory task (Romo task, Figure 1D), where two scalar signals are successively presented, interleaved by a delay period. The task of the network is to report the difference between the values of the two stimuli. Doing so requires two computational variables: one that memorizes the first stimulus, and a second that encodes the difference between the two stimuli. Accordingly, we find that the rank is required to be at least $K = 2$, while having $P = 1$ population is sufficient (Figure 1E). In Figure 1F we analyse the dynamics of a reconstructed network, showing how the two internal variables $\kappa_1$ and $\kappa_2$ implement the two computational variables required for solving the task.

Overall the rank of the network determines the dimensionality of the phase space of the recurrent dynamics, and therefore the number of internal variables available to implement the computation.

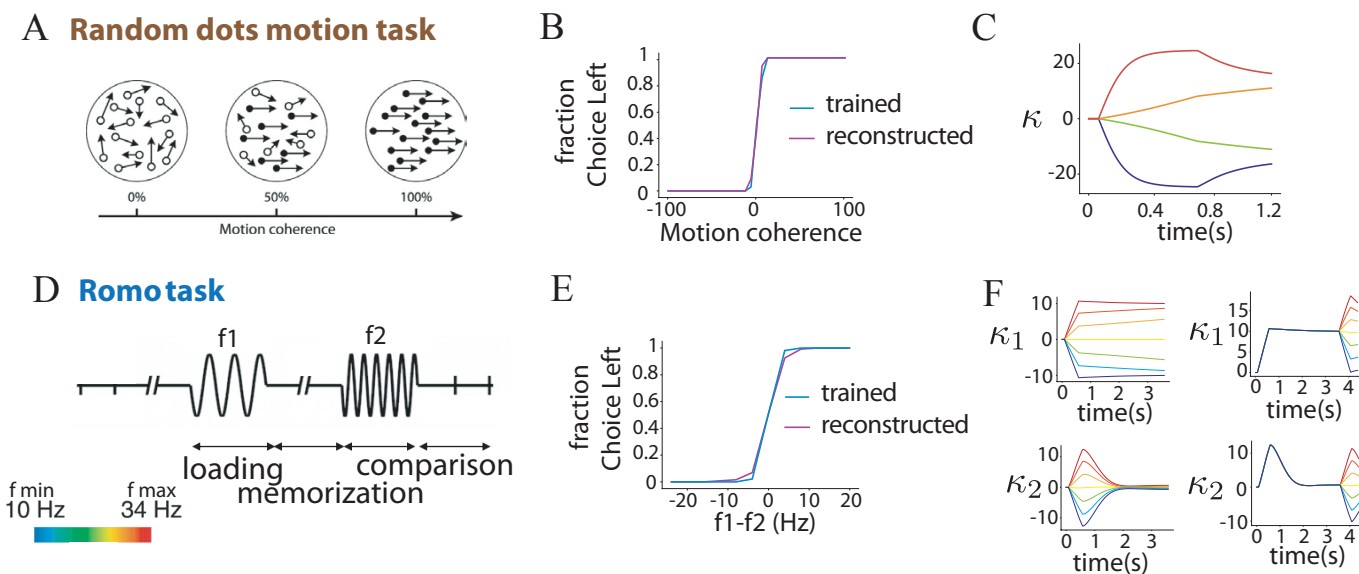

Figure 1: *Computations with neural activities of increasing dimensionality. A. Visual stimuli in the RDM task are dots moving randomly with an overall coherence, either to left or right, that has to be estimated by the subject. The three circles show dots with their instantaneous vector fields for three different trials of decreasing difficulty. B. Psychometric curves from a trained rank-1 network (blue), and from a reconstructed rank-1 network model with one population (magenta). C. Dynamics of the internal variable for four different trials modeling dots moving to the left with high coherence (blue), low coherence (green), to the right with high coherence (red), low coherence (orange). D. Romo task: two stimuli, interleaved by a delay period, have to be compared at the end of the trial. E. Psychometric curves from a trained rank-2 network (blue), and from a reconstructed rank-2 network model with one population (magenta). F. Dynamics of the two internal variables. Left: presentation of the first stimulus and delay, different colors mean different values of the first stimulus $f1$. The first internal variable encodes $f1$ throughout the delay. The second internal variable encodes $f1$ only transiently. Right: whole trial, different colors mean different values of the second stimulus $f2$, $f1 = 34Hz$. After the second stimulus has been presented, $f1 + f2$ is encoded by the first internal variable, $f2$ by the second internal variable, which the read-out can combine to report the comparison $f1 - f2$.*

### 3.2 Multiple populations allow multiple operations to be performed on available internal variables

We now consider a context-dependent perceptual integration task (Mante task, Figure 2A), where two fluctuating scalar signals are presented and the network is asked to integrate only one of the two signals, depending on a contextual cue. The task is a more complex version of RDM, where in addition to an accumulation of evidence mechanism, an attention mechanism is required to flexibly select the integrated signal. We find that a network with a single population is not able to implement this task, whatever the rank. In contrast, a rank-1 network with $P = 2$ is sufficient. The corresponding single internal variable corresponds to integrated evidence, the only computational variable required for solving the task (Figure 2B). Having two populations however allows the network to switch between two operations performed by this variable. This is achieved by reconfiguring the dynamical landscape of the internal variable in a context-dependent manner as illustrated in Figure 2C. Analytical examination of the reconstructed network reveals the underlying mechanism: contextual inputs selectively modulate the gains of populations (Figure 2D), controlling network's functional connectivities, see eq.(4).

More generally, we find that having multiple populations allows the network to flexibly switch between different dynamics of the phase space, and therefore implement several operations on the available internal variables.

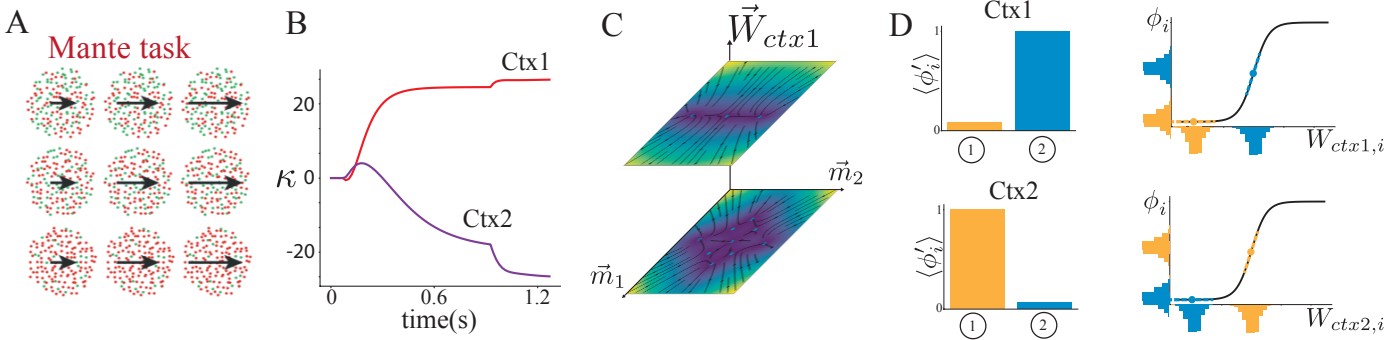

*Figure 2: Flexible computation in multi-population networks. A. Sensory inputs in the Mante task. In addition to exhibiting a coherent motion as in RDM, dots are also colored red or green. A decision about motion should be reported in one context, about color in the other context. B. Dynamics of the internal variable for the same sensory inputs in different contexts. C. Reconfiguration of dynamical landscape by contextual inputs. For visualisation purposes, we consider in this panel a rank-2 network solving the Mante task, and show how a contextual input allows to modify a phase portrait so as to keep only a subset of fixed-points (blue points). D. Contextual inputs modulate selectively the gains of each population (left) by relying on the single-neuron non-linearity (right).*

## 3.3 Multi-tasking networks

Here we draw some perspectives about how these two principles enable the construction of networks performing multiple-tasks. On one hand, increasing dimensionality allows multiple internal variables to process inputs in parallel. On the other hand, increasing the number of populations allows for the selective modulation of more functional connectivities, increasing the flexibility with which the dynamics of an internal variable can be controlled. We illustrate these two principles by constructing networks that perform multiple tasks in parallel, by summing the rank-1 matrix solving RDM, the rank-1 matrix solving Mante task and the rank-2 matrix solving the Romo task to get a single network performing those three tasks simultaneously (Figure 3A) ; and by constructing a rank-1 network of $P$ populations that solves a generalization of the Mante task, with $P$ input streams, with a single internal variable (Figure 3B).

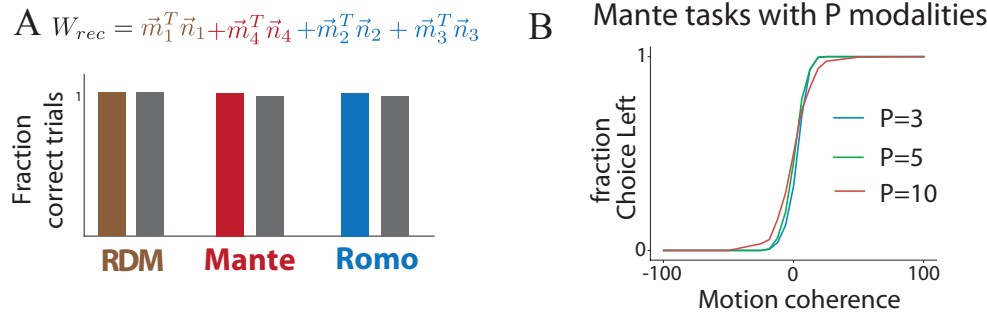

*Figure 3: Multi-tasking networks. A. Performance of networks trained separately on each task (colored) and of the reconstructed multi-tasking network (black). B. Psychometric curves for networks performing the Mante tasks with $P$ input streams, and $P$ corresponding contextual inputs.*

## 4 Conclusion

In this work we have provided an abstract description of computations performed in recurrent neural networks. By focusing on simple tasks this has allowed us to identify the complementary roles of the two important aspects that are dimensionality (section 3.1) and cell classes (section 3.2). Beyond these simple tasks, we have been able to use this understanding to build networks solving multiple tasks (section 3.3), and we expect these principles of neural computation to be important results for the development of procedures aiming at reverse-engineering networks trained on more complex, real-world tasks.

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
