# OpenReview forum: "Disentangling the roles of dimensionality and cell classes in neural computations"
_NeurIPS.cc/2019/Workshop/Neuro_AI — Real Neurons & Hidden Units @ NeurIPS 2019 Poster_

### Official Review · AnonReviewer2 · 2019-09-24
**Clever way to relate a class of recurrent neural networks to tasks in terms of rank of connectivity matrices**

**Clarity:** 4

**Comment:**

Really interesting approach, main limitations are that it's a fairly special case (which I don't find problematic) and that it's a bit preliminary / proof of principle.

**Category:**

AI->Neuro

**Clarity Comment:**

Mostly easy to follow, a little heavy in parts (unavoidably perhaps).

**Evaluation:**

3: Good

**Importance:**

3: Important

**Importance Comment:**

It's nice to be able to relate task complexity to a simple property of connectivity matrices, and to use this to analyse tasks and networks including creating connectivity for multi-task networks. Main issues are (1) it's a very special case and the tasks studied are for the moment very simple, but a lot of promise, (2) not much in the way of actual results, more of a method that could be generalised/applied more widely.

**Intersection:**

4: High

**Intersection Comment:**

Good application of techniques from ML to a neuroscience problem.

**Rigor Comment:**

Seems correct to me, but not enough space to have a lot of detail.

**Technical Rigor:**

3: Convincing

---

### Official Review · AnonReviewer3 · 2019-09-25
**Interesting investigation into the specific utility of cell classes and dimensionality in neural computation**

**Clarity:** 4

**Comment:**

Overall, I thought this was a really great paper. A few comments for improvement:
- There might be a slight discrepancy in how systems neuroscientists talk about 'cell classes' and how this paper does. Generally, cell classes are defined by their functional (or genetic/anatomical) properties, which may be related (or not) to their connectivity to other neurons. Nonetheless, I do think the current study (in investigating how many populations are functionally related to each other) is interesting - I just find the nomenclature, and how it relates to other literature using the same nomenclature, a bit confusing. I think it would be really fascinating to further nail down the link between functionally-defined cell classes (ie cell classes defined the systems neuro way), dynamics, and neural computation.
- The case presented - specifically, the mixture of Gaussians for different cell classes - feels a bit specific. It is nice that there is previous work to build on this, and Gaussians are a great case to start with, but the choice isn't totally motivated and doesn't connect fully with the experimental literature.

**Category:**

Common question to both AI & Neuro

**Clarity Comment:**

This paper was very well-written. It was technical without getting bogged down in detail, and an intuitive description of their simulations and analyses was presented. The authors did a great job of providing motivation for each set of simulations/analyses. Each result was also well-summarized, and I finished reading this paper feeling like I learned something interesting.

**Evaluation:**

4: Very good

**Importance:**

4: Very important

**Importance Comment:**

Lots of recent work, especially in neuroscience, has investigated the relationship between cell class, the dimension of the neural response, and the complexity of the task at hand. As it is difficult to investigate these relationships casually in a real brain, most work on this front has been rather speculative and observational. In probing these properties in artificial systems, the authors make important advances in our understanding of how cell classes and dimensionality underlie computation

**Intersection:**

5: Outstanding

**Intersection Comment:**

While this paper exclusively focused on simulation of artificial neural networks, the general idea and results speak directly to experiments and analyses performed within the experimental/systems neuroscience sphere. I believe that both AI and neuroscience communities will benefit from this work, thus meriting outstanding intersection.

**Rigor Comment:**

From what is presented in the paper, the work seems to be quite rigorous. I appreciate that the authors included the mean field equations for understanding the dynamics of a neural population with a single cell class, and describe roughly how they extend these equations to include multiple classes. The figures presented are consistent with the text, and provide support for their simulations and general scientific argument.

**Technical Rigor:**

4: Very convincing

---

### Official Review · AnonReviewer1 · 2019-09-26
**Techniques to interpret recurrent neural networks trained on systems neuroscience tasks**

**Clarity:** 4

**Comment:**

This paper studies the intersection of several interesting problems in systems neuroscience and neural networks, working in the context of ongoing debates in neuroscience and introducing new techniques in neural networks. Both the rank-restriction and the Gaussian reconstruction could be used in more complex tasks. A sentence or two describing how the rank restriction was imposed, and how the Gaussians were reconstructed, would be welcome.

While the technical exposition was clear, the interpretation with respect to biology and neural networks could be clarified. Biological cell type diversity spans many more parameters than those described by the covariance approach; while the introduction of covariance classes is itself an interesting step, the authors might include some speculation on how this definition of cell class enriches the neuroscientist’s understanding of population dynamics. Similarly, from a mathematical perspective, the authors could clarify why higher rank wouldn't achieve the same goal.

**Category:**

Common question to both AI & Neuro

**Clarity Comment:**

The paper is clearly written. The ultimate interpretation of the results with respect to either neurobiology or neural networks is not entirely clear. (See full comments below.)


**Evaluation:**

4: Very good

**Importance:**

3: Important

**Importance Comment:**

The authors identify a critical issue in the population dynamics view of systems neuroscience: namely, the lack of consideration of cell class. They aim to address this issue by introducing cell classes into rank-constrained recurrent neural networks.


**Intersection:**

4: High

**Intersection Comment:**

The authors address ongoing questions in systems neuroscience and issues of interpretability in conventionally trained neural networks.


**Rigor Comment:**

The authors make effective use of a mix of analytical and computational techniques. Constrained optimization over low-rank weight matrices recovers the low-dimensional structure expected from low-dimensional tasks. The use of well-known tasks in the experimental literature is compelling.


**Technical Rigor:**

4: Very convincing

---

### Decision · Program_Chairs · 2019-10-02

Accept (Poster)